# *Streptomyces* exploration is triggered by fungal interactions and volatile signals

**Stephanie E Jones[1,2], Louis Ho[3], Christiaan A Rees[4,5], Jane E Hill[4,5], Justin R Nodwell[3], Marie A Elliot[1,2]\***

[1]Department of Biology, McMaster University, Hamilton, Canada; [2]Michael G. DeGroote Institute for Infectious Disease Research, McMaster University, Hamilton, Canada; [3]Department Biochemistry, University of Toronto, Toronto, Canada; [4]Geisel School of Medicine, Dartmouth College, Hanover, United States; [5]Thayer School of Engineering, Dartmouth College, Hanover, United States

**Abstract** It has long been thought that the life cycle of *Streptomyces* bacteria encompasses three developmental stages: vegetative hyphae, aerial hyphae and spores. Here, we show interactions between *Streptomyces* and fungi trigger a previously unobserved mode of *Streptomyces* development. We term these *Streptomyces* cells 'explorers', for their ability to adopt a non-branching vegetative hyphal conformation and rapidly transverse solid surfaces. Fungi trigger *Streptomyces* exploratory growth in part by altering the composition of the growth medium, and *Streptomyces* explorer cells can communicate this exploratory behaviour to other physically separated streptomycetes using an airborne volatile organic compound (VOC). These results reveal that interkingdom interactions can trigger novel developmental behaviours in bacteria, here, causing *Streptomyces* to deviate from its classically-defined life cycle. Furthermore, this work provides evidence that VOCs can act as long-range communication signals capable of propagating microbial morphological switches.

**\*For correspondence:** melliot@ mcmaster.ca

**Competing interests:** The authors declare that no competing interests exist.

## Introduction

Our current understanding of microbial growth and development stems largely from investigations conducted using single-species cultures. It is becoming clear, however, that most bacteria and fungi exist as part of larger polymicrobial communities in their natural settings (*Scherlach et al., 2013*; *Traxler and Kolter, 2015*). Microbial behavior is now known to be modulated by neighbouring organisms, where interspecies interactions can have profound and diverse consequences, including modifying virulence of human pathogens (*Peleg et al., 2010*), altering antibiotic resistance profiles of mixed-species biofilms (*Oliveira et al., 2015*), enhancing bacterial growth (*Romano and Kolter, 2005*), and increasing production of specialized metabolites by fungi and bacteria (*Schroeckh et al., 2009*; *Stubbendieck and Straight, 2016*). Consequently, an important next step in advancing our developmental understanding of microbes will be to expand our investigations to include multi-species cultures, and in doing so, unveil new and unexpected microbial growth strategies.

The soil is a heterogeneous environment that is densely populated with bacteria and fungi, and as such, represents an outstanding system in which to study the effects of bacterial-fungal interactions. Within the polymicrobial communities occupying the soil, *Streptomyces* represent the largest genus of the ubiquitous actinomycetes group. These Gram-positive bacteria are renowned for both their complex developmental life cycle (*Elliot et al., 2008*) and their ability to produce an extraordinary range of specialized metabolites having antibiotic, antifungal, antiparasitic, and anticancer properties (*Hopwood, 2007*).

**eLife digest** Soil is home to many bacteria. In fact, soil gets it characteristic 'earthy' smell from a common type of soil bacteria known as *Streptomyces*. Remarkably, *Streptomyces* are also the original sources of most of the antibiotics that are prescribed by doctors to treat bacterial infections. Scientists have been studying *Streptomyces* for over 70 years, and in all this time, there has been unanimous agreement on how these bacteria grow. That is to say that, unlike most other bacteria, *Streptomyces* grow like plants: they don't move, and instead produce spores that are dispersed like seeds. This stationary lifestyle makes these bacteria somewhat vulnerable to predators, and so it is thought that *Streptomyces* make antibiotics to help protect themselves from other bacteria that are able to move around in the soil.

However, this established view of *Streptomyces* growth has now been turned on its head because Jones et al. have discovered that *Streptomyces* bacteria can indeed move when grown in the presence of fungi. Specifically, when a species of *Streptomyces* is grown with yeast, some of the bacteria start to explore their environment, move over top of other bacteria and up hard surfaces to heights that would be the equivalent of humans scaling Mount Everest.

Unexpectedly, Jones et al. also found that these "explorer" *Steptomyces* can communicate with nearby *Streptomyces* bacteria with a perfume-like airborne signal and convince their relatives to begin exploring too. Furthermore, while this volatile signal promotes the growth of *Streptomyces*, it adversely affects other bacteria and makes them sicker such that they are less able to grow and survive.

Together these findings reveal new ways that bacteria and other microbes can interact and communicate with each other. They also emphasise that researchers will need to consider such long-range communication strategies if they hope to better understand microbial communities.

The *Streptomyces* life cycle encompasses three developmental stages (*Figure 1A*). First, a spore germinates to generate one or two germ tubes. These grow by apical tip extension and hyphal branching, ultimately forming a dense vegetative mycelial network that scavenges for nutrients. Second, in response to signals that may be linked to nutrient depletion, non-branching aerial hyphae extend into the air away from the vegetative cells. These aerial hyphae are coated in a hydrophobic sheath that enables escape from the aqueous environment of the vegetative mycelium (*Claessen et al., 2003*; *Elliot et al., 2003*), and their emergence coincides with the onset of specialized metabolism within the vegetative cells (*Kelemen and Buttner, 1998*). Aerial development requires the activity of the '*bld*' gene products, where mutations in these genes result in colonies lacking the fuzzy/hydrophobic characteristics of wild type. The final developmental stage involves the differentiation of aerial hyphae into spores through a synchronous cell division and cell maturation event. This process is governed by the *whi* (for 'white') gene products, whose mutants fail to form mature, pigmented spores (*McCormick and Flärdh, 2012*). In addition to being highly stress-resistant, spores also provide a means of dispersing *Streptomyces* to new environments, as all characterized *Streptomyces* cell types are non-motile.

In this work, we identify a novel interaction between *Streptomyces venezuelae* and fungal microbes that induces a previously unknown mode of bacterial growth. We refer to this as 'exploratory growth', whereby cells adopt a non-branching vegetative hyphal conformation that can rapidly traverse both biotic and abiotic surfaces. We show that part of the mechanism by which fungi induce exploratory growth involves glucose depletion of the growth medium. Remarkably, this novel mode of growth can be communicated to other – physically separated – streptomycetes through a volatile compound. Volatile signalling further alters cell propagation and survival of other bacteria.

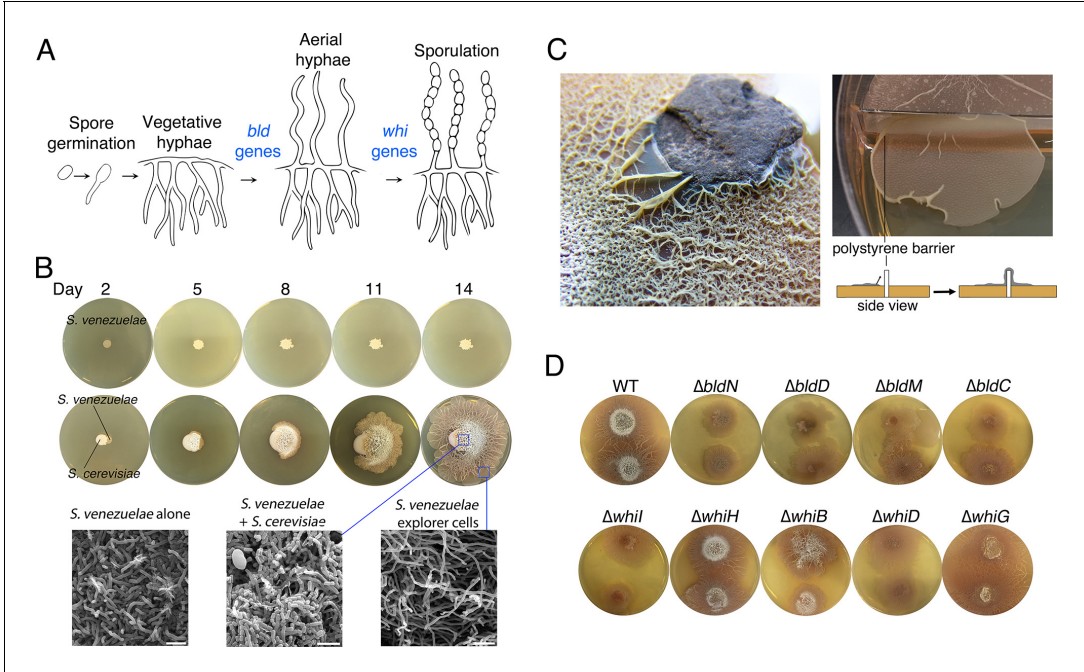

**Figure 1.** Physical association with yeast triggers *Streptomyces* exploratory behaviour. (A) Developmental life cycle of *Streptomyces*. Germ tubes emerge from a single spore, and grow by apical tip extension and hyphal branching, forming a dense network of branching vegetative hyphae. In response to unknown signals, non-branching aerial hyphae coated in a hydrophobic sheath, escape into the air. Aerial hyphae differentiate into chains of dormant, stress-resistant non-motile spores. The *bld* gene products are required for the transition from vegetative growth to aerial hyphae formation, while the *whi* gene products are required for the differentiation of aerial hyphae into spore chains. (B) *S. venezuelae* grown alone (top row) and beside *S. cerevisiae* (middle row) on YPD (yeast extract-peptone-dextrose) medium over 14 days. Bottom panels: scanning electron micrographs of *S. venezuelae* grown alone (left), *S. venezuelae* on *S. cerevisiae* (middle), and *S. venezuelae* beside *S. cerevisiae* (right) for 14 days on YPD agar medium. White bars: 5 μm. (C) *S. venezuelae* explorer cells growing up a rock embedded in agar (left), and over a polystyrene barrier within a divided petri dish (right, and schematic below). (D) *S. venezuelae* wild type and developmental mutants grown beside *S. cerevisiae* on YPD agar medium for 14 days. Top: *S. cerevisiae*, together with wild type and Δ*bld* mutant strains (*bld* mutants cannot raise aerial hyphae and sporulate). Bottom: *S. cerevisiae* grown next to Δ*whi* mutant strains (*whi* mutants can raise aerial hyphae but fail to sporulate).

The following figure supplements are available for figure 1:

**Figure supplement 1.** Explorer cells are hydrophilic.

**Figure supplement 2.** Phylogeny of exploratory streptomycetes.

**Figure supplement 3.** *S. venezuelae* grown beside diverse yeast strains.

## Results

### Physical association with yeast stimulates *Streptomyces* exploratory behaviour

To explore interactions between *Streptomyces* and fungi, we cultured *Streptomyces venezuelae* alone or beside the yeast *Saccharomyces cerevisiae* on solid agar (*Figure 1B*), and incubated these cultures for 14 days. As expected, during this time *S. venezuelae* on its own formed a colony of normal size. In contrast, when *S. venezuelae* was grown beside *S. cerevisiae*, its growth was radically different. During the first five days, the cells appeared to consume *S. cerevisiae*, before initiating a rapid outgrowth that led to *S. venezuelae* colonizing the entire surface of a 10 cm agar plate after 14 days. Remarkably, growth did not cease when physical obstructions were encountered: *S. venezuelae* cells were able to spread over rocks and polystyrene barriers (*Figure 1C*).

To gain insight into this phenomenon, we visualized the leading edge of the rapidly migrating *S. venezuelae* cells (*Video 1*). We found it initially progressed at a rate of ~1.5 µm/min. This is an order of magnitude faster than would be explained by growth alone, given that hyphal tip extension has been calculated to occur at a rate of 0.13 µm/min (*Richards et al., 2012*). We refer to this rapid movement as 'exploratory growth', and these spreading cells as 'explorers', based on their ability to effectively transverse both biotic and abiotic surfaces. To further investigate the morphology of these explorer cells, we used scanning electron microscopy (SEM) to visualize *S. venezuelae* grown alone, *S. venezuelae* at the yeast interface, and *S. venezuelae* explorer cells, after 14 days of growth (*Figure 1B*). We found *S. venezuelae* alone grew vegetatively, albeit without any obvious branches (branching vegetative cells were observed during growth on other media types, as expected), whereas *S. venezuelae* growing on *S. cerevisiae* raised aerial hyphae and sporulated. Microscopic analysis of explorer cells revealed that they failed to branch and were reminiscent of aerial hyphae. Unlike aerial hyphae, however, these filaments were hydrophilic, based on their inability to repel aqueous solutions (*Figure 1—figure supplement 1*).

To determine whether exploratory growth required classic developmental regulators (the *bld* and *whi* gene products), we grew a suite of *S. venezuelae* developmental mutants beside *S. cerevisiae* to evaluate whether these mutations impacted colony spreading (*Figure 1D*). Four *S. venezuelae bld* mutants (*bldC,D,M,N*) and five *S. venezuelae whi* mutants (*whiB,D,G,H,I*) were inoculated beside *S. cerevisiae*. Unexpectedly, all developmental mutant strains displayed a similar exploratory behaviour as wild type after 14 days, although the *bldN* mutant exhibited slower exploration than the other strains. The mutant strains did, however, differ in their growth on yeast, with the *bld* mutants failing to raise aerial hyphae, and the *whi* mutants failing to sporulate. This demonstrated that exploratory growth was distinct from the canonical *Streptomyces* life cycle, and represented a new form of growth for these bacteria.

To determine whether this exploratory behaviour was unique to *S. venezuelae*, we inoculated other commonly studied streptomycetes beside *S. cerevisiae*. We found that well-studied *Streptomyces* species, including *S. coelicolor*, *S. avermitilis*, *S. griseus*, and *S. lividans,* failed to exhibit an analogous spreading behaviour when plated next to *S. cerevisiae*. We next tested 200 wild *Streptomyces* isolates, growing each beside *S. cerevisiae*. Of these, 19 strains (~10%) exhibited exploratory growth similar to *S. venezuelae*. To determine whether this behaviour was confined to a particular *Streptomyces* lineage, we performed a phylogenetic analysis of these explorer-competent strains using *rpoB* sequences, and included non-exploratory model *Streptomyces* species as outgroups (*Figure 1—figure supplement 2*). We found *S. venezuelae* and these wild *Streptomyces* did not form a monophyletic group, suggesting that exploratory growth is wide-spread in the streptomycetes.

We next sought to determine whether *Streptomyces* exploratory behaviour could be triggered by other fungi. *S. venezuelae* was inoculated beside laboratory strains of *Candida albicans*, *Candida parapsilosis,* and *Crypotococcus neoformans*, and beside wild soil isolates of *S. cerevisiae, Zygosaccharomyces florentinus, Saccharomyces castellii, Pichia fermentans* and *Debaryomyces hansenii* (*Figure 1—figure supplement 3*). We observed that all species, apart from *C. neoformans* and *P. fermentans*, induced *S. venezuelae* exploratory behaviour. This indicated that a broad range of microbial fungi could trigger exploratory growth.

## The yeast TCA cycle must be intact to stimulate *S. venezuelae* exploratory behaviour

To understand how fungi could stimulate exploration, we took advantage of an *S. cerevisiae* haploid knockout collection containing 4309 individual knockout strains. Each *S. cerevisiae* mutant was pinned adjacent to *S. venezuelae* and after 10 days, *S. venezuelae* exploratory growth was assessed. We identified 16 mutants that were unable to promote *S. venezuelae* exploration (*Figure 2A*). Of these, 13 had

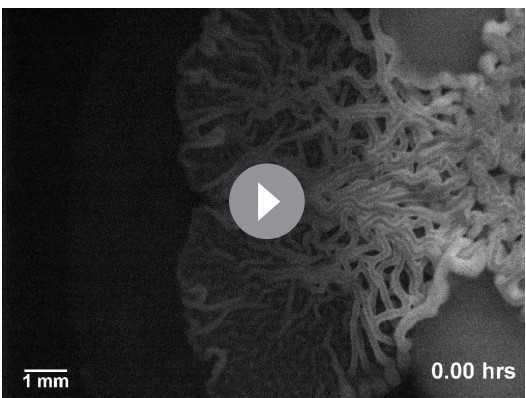

**Video 1.** Leading edge of *S. venezuelae* explorer cells over a 17 hr time course.

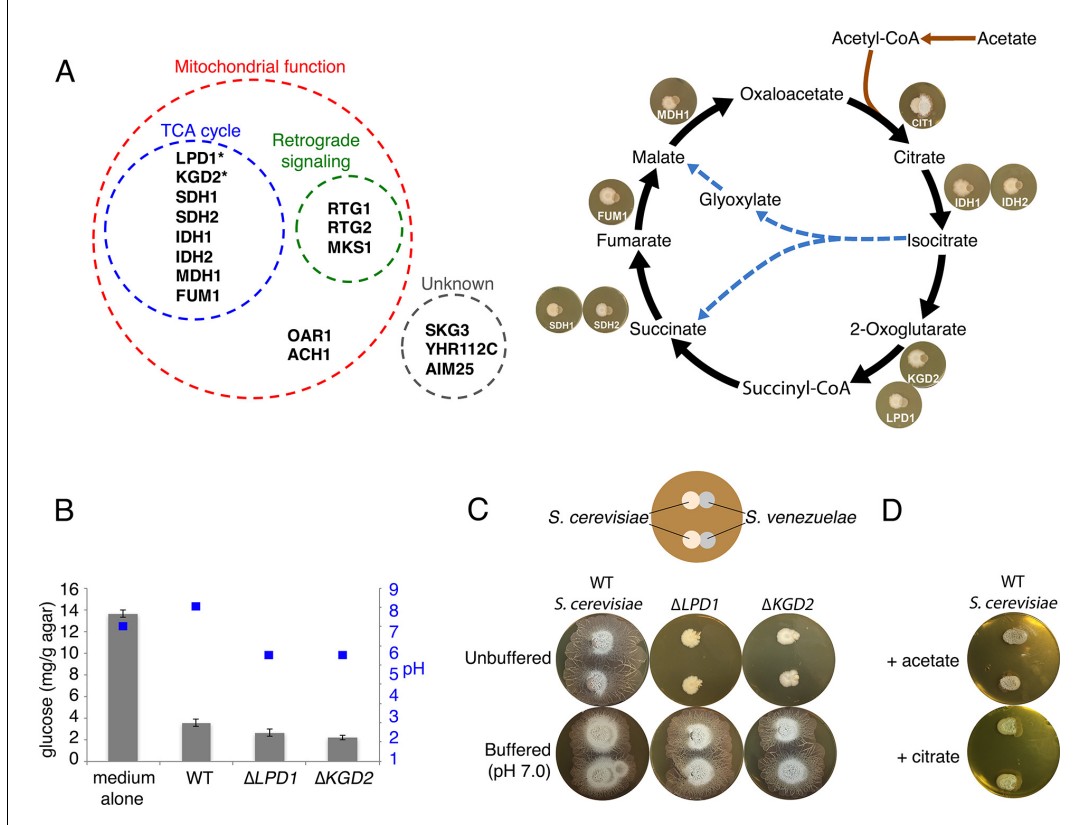

**Figure 2.** Yeast stimulates *S. venezuelae* exploratory growth by consuming glucose and inhibits it by acidifying the medium. (**A**) *S. cerevisiae* mutants that fail to stimulate *S. venezuelae* exploratory growth. Left: functional grouping of the exploration-deficient *S. cerevisiae* mutations. Asterisks indicate genes also identified in *C. albicans* as affecting *S. venezuelae* exploratory growth. Right: Mutations in *S. cerevisiae* TCA cycle-associated genes affect exploration after citrate production. For each interaction, the indicated *S. cerevisiae* mutant was grown beside wild type *S. venezuelae* for seven days on YPD agar medium. (**B**) Glucose concentration and pH associated with wild type and mutant *S. cerevisiae* strains grown on YPD agar medium. Glucose concentrations (grey bars) and pH (blue squares) were measured from medium alone, and beneath wild type, Δ*LPD1* or Δ*KGD2 S. cerevisiae* strains grown on YPD medium for seven days. All values represent the mean ± standard error for four replicates. (**C**) Top: schematic of the experimental set up, with *S. cerevisiae* grown to the left of *S. venezuelae* on YPD medium. Two replicates are grown on each agar plate. Bottom: wild type, Δ*LPD1*, and Δ*KGD2 S. cerevisiae* strains grown for 14 days beside wild type *S. venezuelae* on unbuffered YPD agar and YPD agar buffered to pH 7.0 with MOPS. (**D**) Wild type *S. cerevisiae* spotted beside wild type *S. venezuelae* and grown for 14 days on YPD agar medium plates supplemented with acetate or citrate, each buffered to pH 5.5.

The following figure supplements are available for figure 2:

**Figure supplement 1.** *C. albicans* gene mutations that affect *S. venezuelae* exploratory growth.

**Figure supplement 2.** *S. venezuelae* grown alone on glucose-deficient medium exhibits similar exploratory growth to *S. venezuelae* growing next to yeast on glucose medium.

mutations affecting mitochondrial function, including eight in genes coding for enzymes in the tricarboxylic acid (TCA) cycle (*Figure 2A*), three in genes whose products contribute to the mitochondrial retrograde signalling pathway, as well as two whose products are involved in mitochondrial metabolism.

We confirmed these mutant effects using *Candida albicans* strains carrying tetracycline-repressible haploid mutations (*Figure 2—figure supplement 1*). We grew four mutant strains adjacent to *S. venezuelae*, and found that two of them, Δ*LPD1* and Δ*KGD2,* also failed to stimulate *S. venezuelae* exploratory behaviour. As the products of these two genes act in the TCA cycle (*Figure 2A*), these data collectively suggest that fungal respiration, and in particular TCA cycle function, influences exploratory growth in *S. venezuelae*.

## Exploration is glucose-repressible and pH-dependent

In considering how TCA cycle defects could influence *S. venezuelae* behaviour, we hypothesized that glucose uptake and/or consumption might play a role. We measured glucose levels of a YPD agar control, and compared this with YPD agar underneath *S. cerevisiae*. Uninoculated medium had 3.8 times as much glucose as *S. cerevisiae*-associated agar (*Figure 2B*), confirming that *S. cerevisiae* consumed glucose during growth on YPD agar. This suggested that either glucose depletion by yeast, or some product of glucose metabolism, may trigger *S. venezuelae* exploratory growth.

To test these possibilities, we first asked whether exploratory growth could be triggered by lowering glucose concentrations. We plated *S. venezuelae* on YP (yeast extract-peptone) in the presence (G+) and absence (G−) of glucose (*Figure 2—figure supplement 2*). After 10 days, we found growth on G− medium permitted *S. venezuelae* exploration, irrespective of whether yeast was present. This implied that glucose repressed exploratory growth.

We also tested glucose consumption by the *S. cerevisiae LPD1* and *KGD2* mutants. The products of these genes, along with that of *KGD1*, comprise the 2-oxoglutarate dehydrogenase complex responsible for converting 2-oxoglutarate into succinyl-CoA in the TCA cycle (*Przybyla-Zawislak et al., 1999*) (*Figure 2A*). We found wild type, ΔLPD1 and ΔKGD2 *S. cerevisiae* strains consumed similar levels of glucose (*Figure 2B*), suggesting that other factors must be inhibiting *S. venezuelae* exploration when grown adjacent to these TCA cycle mutants.

All TCA cycle-associated *S. cerevisiae* mutants that failed to stimulate *S. venezuelae* exploratory behaviour were blocked after the production of citrate in the TCA cycle (*Figure 2A*). We hypothesized that this disruption might result in an accumulation of organic acids, and that *S. cerevisiae* mutants secreted these acids to maintain a neutral intracellular pH. We measured the pH of wild type, ΔLPD1, and ΔKGD2 strains when grown on YPD (G+) agar, and found wild type *S. cerevisiae* raised the agar pH from 7.0 to 7.5, whereas both TCA cycle mutants lowered the agar pH to 5.5 (*Figure 2B*).

To test whether acid secretion by the *S. cerevisiae LPD1* and *KGD2* mutants prevented *S. venezuelae* exploratory growth, the two mutants were grown beside *S. venezuelae* on non-buffered YPD agar, and equivalent medium buffered to pH 7.0 (*Figure 2C*). After 14 days growth on non-buffered plates, the *S. cerevisiae* mutants failed to stimulate *S. venezuelae* exploratory behaviour, whereas the same strains on buffered agar – which would counter the pH-lowering effects of the secreted acids – could now promote *S. venezuelae* exploration. To further verify this pH-dependent effect, we grew wild type *S. cerevisiae* beside *S. venezuelae* on YPD agar supplemented with citrate or acetate (*Figure 2D*). We found that after 14 days, *S. venezuelae* spreading was inhibited, confirming that secreted acids inhibited *S. venezuelae* exploration.

Collectively, these results suggested that *S. venezuelae* exploratory growth is a glucose- and acid-repressible phenomenon. Consistent with these observations, we also determined that *S. venezuelae* exploration was associated with a significant rise in pH: as *S. venezuelae* consumed the yeast, the medium pH rose from 7.0 to 8.0, and once *S. venezuelae* exploratory growth initiated (day 5), the pH rose further to 9.5 (*Figure 3A*). This increase in pH was also observed for *S. venezuelae* grown on G- medium (in the absence of yeast) (*Figure 3—figure supplement 1*), suggesting that the rise in pH was mediated by the *Streptomyces* cells. To determine whether high pH was sufficient to promote exploration, we inoculated *S. venezuelae* cells on YPD agar medium buffered to pH 9.0. Exploration was not induced under these growth conditions (*Figure 3—figure supplement 2*). These data indicated that alkaline conditions were important but not sufficient for exploration, and further suggested that an adaptation phase was required during the transition to exploratory growth.

## *S. venezuelae* exploration requires an alkaline stress response

To investigate the genetic basis for this phenomenon we employed chemical mutagenesis, and screened for *S. venezuelae* mutants that failed to display exploratory behaviour when grown on G-medium (where yeast is not required) (*Figure 3B*). Candidate non-spreading mutant colonies were identified, and were tested in association with *S. cerevisiae* on YPD (G+) medium to confirm their inability to spread. Of the 48 exploration-defective mutants identified on G− medium, only three were also unable to spread when grown on YPD medium beside *S. cerevisiae*. This indicated that exploratory growth on G− agar may have distinct genetic requirements from exploratory growth on YPD (G+) medium.

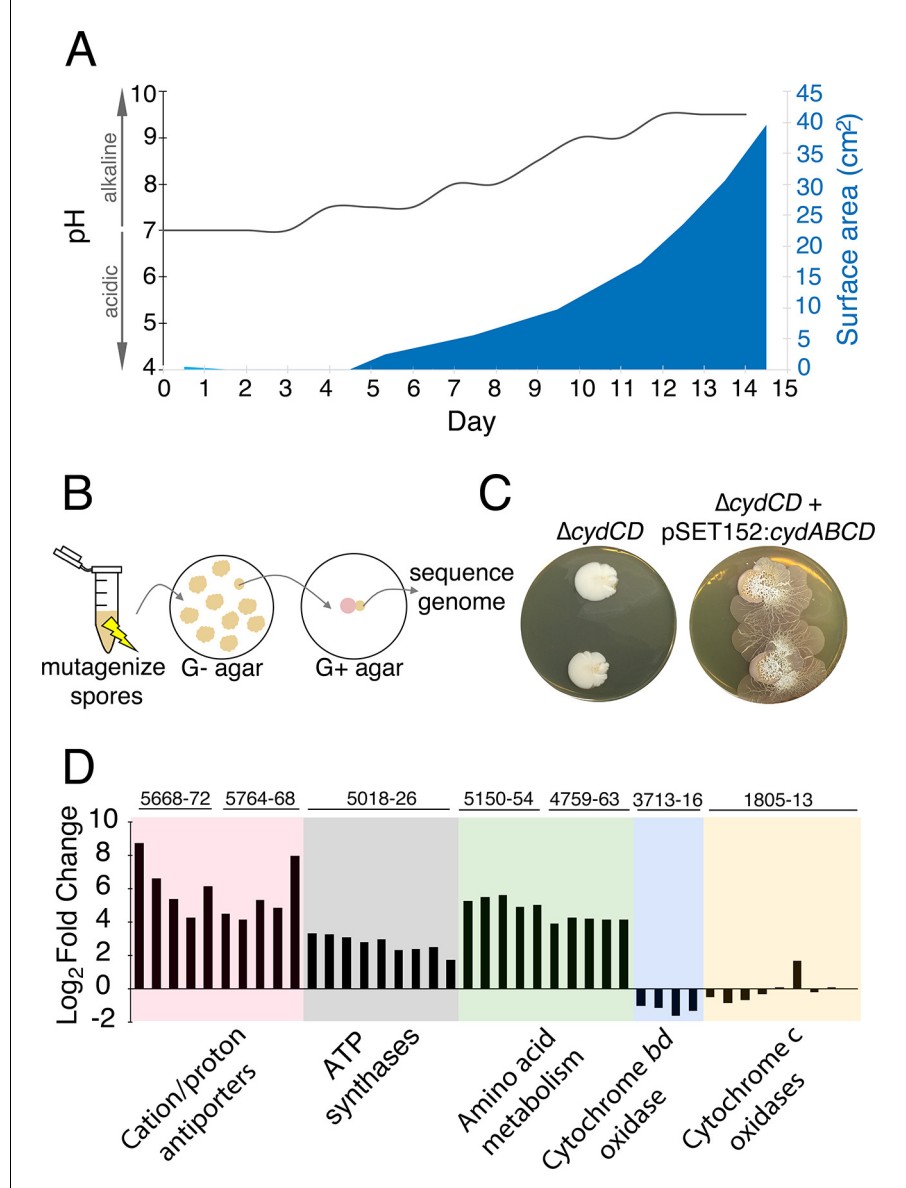

**Figure 3.** The alkaline stress response is associated with *S. venezuelae* exploratory behaviour. (**A**) The surface area and medium pH associated with *S. venezuelae* explorer cells beside *S. cerevisiae* on YPD agar were measured and plotted every day for 14 days. (**B**) Schematic of the method used to identify genes required for *S. venezuelae* exploratory growth. *S. venezuelae* spores were subject to chemical mutagenesis, then screened on G- agar (no glucose, exploration-permissive without *S. cerevisiae*) for a lack of exploratory growth. Static colonies (beige) were grown beside *S. cerevisiae* (pink) on YPD medium to confirm a lack of exploratory growth. Genomic DNA was isolated from strains unable to initiate exploratory growth on G- agar, and when inoculated beside *S. cerevisiae* on YPD medium. Whole genome sequencing was performed to identify mutations responsible for the lack of exploratory growth. (**C**) Morphology of a mutant cytochrome *bd* oxidase *S. venezuelae* strain (Δ*cydCD*) and the corresponding complemented strain grown on YPD agar for 14 days. (**D**) Transcript levels for alkaline stress-responsive genes in *S. venezuelae* explorer cells (grown beside *S. cerevisiae* on YPD medium), divided by levels for non-exploratory *S. venezuelae* cells (grown alone on YPD medium). Transcript levels were normalized and differential expression was log2-transformed. The associated *sven* gene numbers are shown above the bar graphs.

The following figure supplements are available for figure 3:

**Figure supplement 1.** *S. venezuelae* grown alone raises the pH of glucose-deficient medium.

*Figure 3 continued on next page*

*Figure 3 continued*

**Figure supplement 2.** High pH alone does not stimulate *S. venezuelae* exploration.
**Figure supplement 3.** Complementation of explorer mutant phenotypes.

We sequenced the genomes of wild type *S. venezuelae* and the three non-spreading mutants of interest (those unable to spread on both G- medium alone and YPD (G+) medium beside *S. cerevisiae*). Each mutant harbored point mutations in the *sven_3713-3716* operon. This operon is predicted to encode subunits of the cytochrome *bd* oxidase complex (*cydA/sven_3713* and *cydB/ sven_3714*), along with an ABC transporter required for cytochrome assembly (*cydCD/sven_3715*) (*Brekasis and Paget, 2003*). One strain had a mutation in *sven_3715* (H673Y), while the other two strains had mutations in *sven_3713* (Q186stop) and were likely clonal. To ensure that these mutations were responsible for the exploration-defective phenotype, we complemented the exploratory growth defect in each mutant with a cosmid carrying an intact *cydABCD* operon, and confirmed that exploration was restored (*Figure 3—figure supplement 3*). We also deleted *cydCD* in a wild type *S. venezuelae* background, and confirmed that this strain was unable to initiate exploration when grown beside *S. cerevisiae*. As before, spreading could be restored to the mutant after introducing *cydABCD* on an integrating plasmid vector (*Figure 3C*). These data indicated that the cytochrome *bd* oxidase complex was essential for *S. venezuelae* exploration.

*S. venezuelae,* like many other bacteria, encodes two cytochrome oxidase complexes. The cytochrome *bd* oxidase catalyzes terminal electron transfer without a concomitant pumping of protons across the membrane, while the cytochrome $bc_1$-$aa_3$ complex requires proton transfer from the cytoplasm. The cytochrome *bd* oxidase functions as part of the alkaline stress response in other bacteria (*Krulwich et al., 2011*). As we had established that alkaline conditions were a prerequisite for *S. venezuelae* exploration, we questioned whether other alkaline stress-responsive genes might be associated with exploratory growth. Using RNA-sequencing (RNA-seq), we examined the transcription profiles of *S. venezuelae* alone, compared with *S. venezuelae* exploratory cells grown beside *S. cerevisiae* on YPD medium (*Figure 3D*). The five gene clusters mostly highly upregulated in *S. venezuelae* explorer cells encoded the ATP synthase complex (*sven_5018-26*; 7.6-fold increase relative to non-spreading), two predicted cation/proton antiporter complexes (95.9- and 85.3-fold increase relative to non-spreading for *sven_5668-72* and *sven_5764-68*, respectively), and two peptide transporters (17.4- and 38.3-fold increase relative to non-spreading for *sven_4759-63* and *sven_5150-54*, respectively) (*Figure 3D*).

Higher expression of the cation/proton antiporters, alongside increased ATP synthesis, would be expected to enhance proton uptake into the cell; equivalent genes are upregulated as part of the alkaline stress response in other bacteria (*Krulwich et al., 2011*). Amino acid catabolism is also upregulated under alkaline growth conditions in other bacteria (*Padan et al., 2005*). Given the dramatically increased expression of the peptide transporters, we confirmed that exploratory growth required an amino acid source (*Supplementary file 1*). Collectively, these results suggest that exploration is coupled with a metabolic reprogramming that permits robust growth under highly alkaline conditions.

## *S. venezuelae* explorer cells alkalinize the medium using an airborne volatile organic compound

*S. venezuelae* exploration is associated with high pH conditions, and our data suggested this rise in pH was promoted by *S. venezuelae* itself. We hypothesized that this pH effect could be mediated either through the secretion of diffusible basic compounds, or through the release of volatile organic compounds (VOCs). To differentiate between these possibilities, we set up a two-compartment petri plate assay, where *S. venezuelae* was grown beside *S. cerevisiae* on YPD agar in one compartment, while the adjacent compartment contained uninoculated YPD agar (*Figure 4A*). As a negative control, we set up an equivalent set of plates, only with *S. venezuelae* alone (no yeast) on YPD agar in the first compartment. In each case, the two compartments were separated by a polystyrene barrier. After 10 days, we measured the pH of the uninoculated YPD compartment, and found the

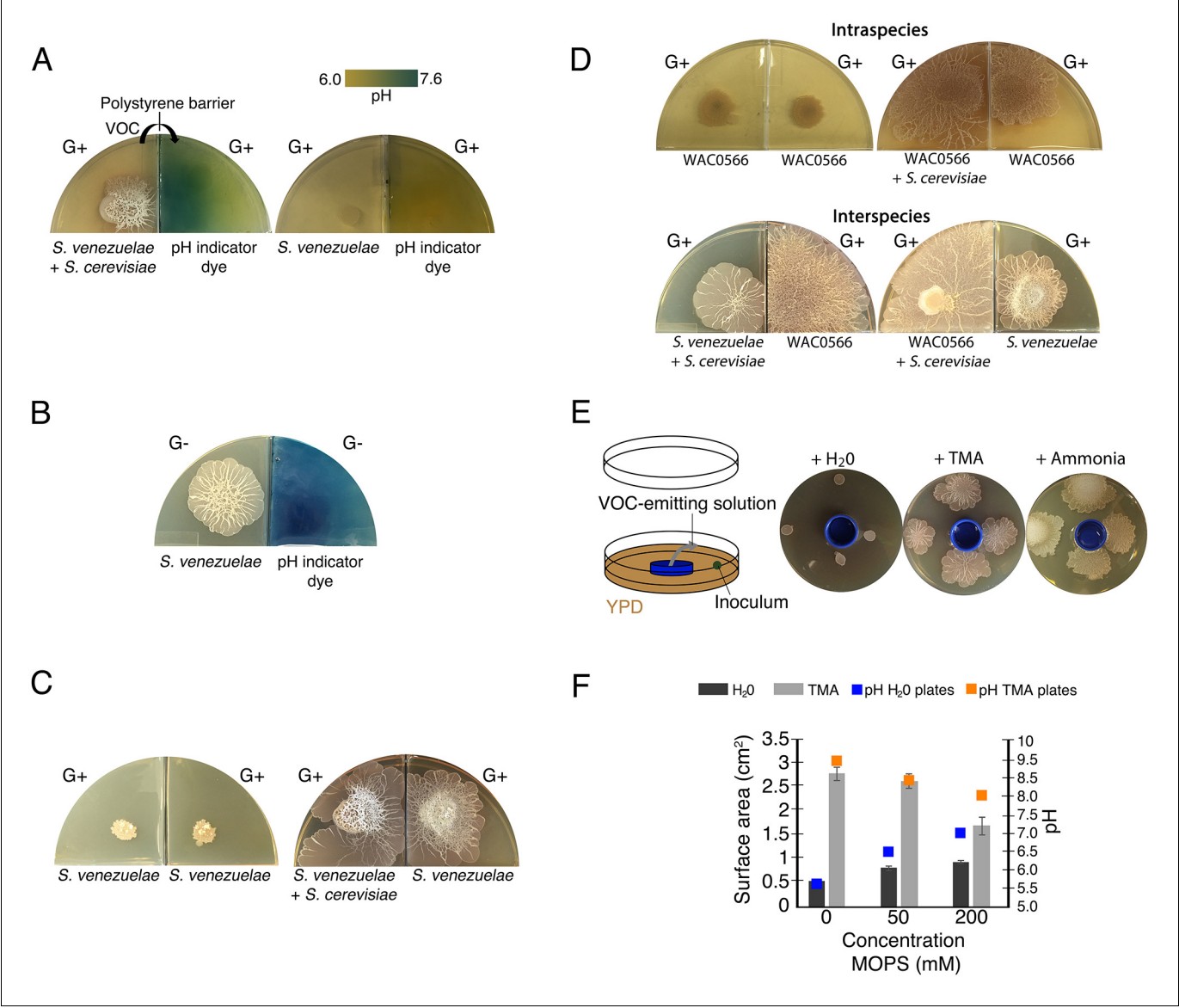

**Figure 4.** Volatile organic compounds released by *S. venezuelae* raise the medium pH and induce exploratory growth in physically separated *Streptomyces*. (**A**) Effect of *S. venezuelae* explorer cells on pH of physically separated medium. Each compartment is separated by a polystyrene barrier. *S. venezuelae* and *S. cerevisiae* were grown in the left compartment of one plate (left), while *S. venezuelae* alone was grown in the left compartment of the other plate (right). After 10 days, bromothymol blue pH indicator dye was spread on the agar in the right compartment of each plate. Blue indicates VOC-induced alkalinity. (**B**) *S. venezuelae* was grown alone on YP (G- agar) in the left compartment, while the right compartment contained uninoculated YP (G-) agar. After seven days, the same pH indicator dye as in **Figure 4A** was spread over the agar in the right compartment. Blue represents a rise in pH above 7.6. (**C**) Left: *S. venezuelae* alone was inoculated in each compartment. Right: *S. venezuelae* was grown beside *S. cerevisiae* in the left compartment, and *S. venezuelae* alone was grown in the right compartment. All strains were grown on YPD (G+) agar medium for 10 days. (**D**) Top left: Wild *Streptomyces* isolate WAC0566 was grown alone in each compartment. Top right: WAC0566 was grown beside *S. cerevisiae* in the left compartment, and grown alone in the right compartment. Bottom left: *S. venezuelae* was grown beside *S. cerevisiae* in the left compartment, and WAC0566 was grown alone in the right compartment. Bottom right: WAC0566 was grown beside *S. cerevisiae* in the left compartment, while *S. venezuelae* was grown alone in the right compartment. All strains were cultured on YPD (G+) agar medium for 10 days. (**E**) Schematic of the plate-based assay used to assess the effects of volatile-emitting solutions (and controls) on nearby *Streptomyces* colonies. $H_2O$, TMA, or ammonia solutions were placed in a blue plastic dish, and *S. venezuelae* was spotted around each dish on YPD medium. Plates were incubated at room temperature for seven days. (**F**) Surface area and pH of *S. venezuelae* colonies grown on YPD medium around small dishes containing $H_2O$ or TMA solutions, as shown in **Figure 4E**. *S. venezuelae* was grown at room temperature for seven days on either unbuffered YPD medium or YPD medium buffered to pH 7.0 using MOPS. All values represent the mean ± standard error for four replicates.

*Figure 4 continued on next page*

*Figure 4 continued*

The following figure supplements are available for figure 4:

**Figure supplement 1.** The *S. venezuelae cydCD* mutant strain can explore in response to volatile signals produced by neighbouring explorer cells.

**Figure supplement 2.** Wild explorer *Streptomyces* species promote exploration in *S. venezuelae* using volatile signals.

**Figure supplement 3.** The VOC produced by *S. venezuelae* explorer cells can be produced by liquid-grown (G-) *S. venezuelae* and WAC0566 cultures.

compartment adjacent to *S. venezuelae* alone remained at pH 7.0, whereas the one adjacent to *S. venezuelae* explorer cells had risen from pH 7.0 to 9.5, indicating the explorer cells produced a basic VOC (*Figure 4A*).

To verify that the VOC was produced by *S. venezuelae* explorers and not by *S. cerevisiae*, we repeated the two-compartment assay with *S. venezuelae* grown alone on G- agar, a condition that also induced exploratory behaviour. We found that *S. venezuelae* growing alone on G- agar could alkalinize the adjacent YPD compartment. This confirmed that a basic VOC was produced by *S. venezuelae* explorer cells (*Figure 4B*).

## *S. venezuelae* exploratory cells use VOCs to induce exploration in other streptomycetes at a distance

Bacterial VOCs can influence a wide range of cellular behaviours. To determine whether the VOC produced by explorer cells represented an exploration-promoting signal for physically separated *Streptomyces* colonies, we leveraged our two-compartment assay, inoculating one with *S. venezuelae* beside *S. cerevisiae* on YPD agar, and the adjacent compartment with *S. venezuelae* on the same medium (a condition where exploration by *S. venezuelae* otherwise requires yeast association). As expected, after 10 days, the *S. cerevisiae*-associated cells were actively spreading. Remarkably, the adjacent *S. venezuelae* cells (in the absence of yeast) had also initiated exploratory growth (*Figure 4C*). As a negative control, *S. venezuelae* alone was grown in both compartments on YPD agar; spreading was not observed for cells grown in either compartment after 10 days (*Figure 4C*). These data implied that *S. venezuelae* explorer cells released a VOC that effectively promoted exploratory growth in distantly located *S. venezuelae* cells. We tested whether our exploration-deficient *cydCD* mutant was able to respond to this VOC, and observed that despite its inability to explore when grown next to yeast, this mutant was capable of exploration when stimulated by neighbouring explorer cells (*Figure 4—figure supplement 1*).

To determine whether *S. venezuelae* explorers used VOCs to potentiate exploration in other species, we again used our two-compartment assay. We cultured *S. venezuelae* with *S. cerevisiae* in one compartment, and tested whether these cells could stimulate exploratory growth of the wild *Streptomyces* isolate WAC0566 in the adjacent compartment (*Figure 4D*) (WAC0566 initiates exploratory growth when cultured next to yeast, but fails to spread on its own; *Figure 4D*). Negative control plates were set up in the same way as before, with WAC0566 alone in both compartments. After 10 days, WAC0566 grown adjacent to *S. venezuelae* explorers initiated exploratory growth, and this was not seen for the negative control (*Figure 4D*). This indicated that exploratory growth could be communicated to unrelated streptomycetes.

We tested the volatile-mediated communication between these strains in a reciprocal experiment, and found that *S. venezuelae* exploration could also be stimulated by a VOC produced by yeast-associated WAC0566 (*Figure 4D*). This inter-species promotion of *S. venezuelae* exploration was observed for at least 13 other wild *Streptomyces* strains (*Figure 4—figure supplement 2*). Importantly, VOC communication of exploratory growth was confined to those species with exploratory capabilities (*S. coelicolor* failed to respond to the VOC elicitor).

## The VOC trimethylamine stimulates *Streptomyces* exploratory behaviour

We determined that the exploration-promoting VOC could be produced by liquid-grown (G-) cultures, and that it stimulated exploratory growth by both *S. venezuelae* and WAC0566 (*Figure 4—*

*figure supplement 3*). To rule out the possibility that any liquid-grown culture could promote exploration, we also grew *S. venezuelae* and WAC0566 in YPD (G+) liquid medium, and found these cultures were unable to stimulate exploration. This suggested that VOC production was glucose-repressible, and its production correlated with growth conditions that promoted exploration.

To determine the identity of the VOC, we grew *S. venezuelae* and WAC0566 in G+ and G- liquid culture for three days. We collected the supernatants of each culture, and assayed them using two-dimensional gas chromatography time-of-flight mass spectrometry (GC×GC-TOFMS). From this, 1400 unique compounds were identified. To determine which compound(s) were responsible for promoting exploration, we applied a stringent filter, requiring the compound(s) to be: (i) present in at least 50% of *S. venezuelae* and WAC0566 exploration-inducing (G-) cultures; (ii) present in at least 10-fold greater abundance in exploration-inducing (G-) cultures versus static (G+) cultures; and (iii) have at least a 60% similarity score to known compounds in the 2011 National Institute of Standards and Technology (NIST) Mass Spectral Library. We arrived at a list of 21 candidate compounds (*Supplementary file 1*). Of these, 12 were not detected in the negative controls (G+ cultures). Within this group of 12, only four were detected in 100% of *S. venezuelae* and WAC0566 exploration-promoting cultures: trimethylamine (TMA), thiocyanic acid, 6-methyl-5-hepten-2-one, and 2-acetylthiazole. Notably, TMA was >10 fold more abundant than the other three compounds, and thus we focussed our initial investigations on this molecule.

TMA is a volatile nitrogen-containing metabolite with a high pKa (9.81). As we knew *S. venezuelae* produced a basic VOC, we hypothesized that TMA was responsible for promoting exploration. To test this possibility,we placed commercially-available TMA in a small plastic container at the centre of a YPD (G+) agar plate, and then inoculated *S. venezuelae* at defined positions around this container (*Figure 4E*). After seven days, *S. venezuelae* cultured adjacent to the TMA-emitting solutions had initiated exploratory growth, while those grown next to a water-containing control failed to spread. This implied that TMA was the VOC used by *S. venezuelae* and WAC0566 to elicit exploratory growth.

TMA production is not well understood, although recent work has revealed two mechanisms by which it can be generated from quaternary amines. *Acinetobacter* sp. employ a carnitine oxygenase (product of the *cntAB* gene cluster) in converting L-carnitine into TMA (*Zhu et al., 2014*), while *Desulfovibrio desulfuricans* converts choline into TMA using a choline-trimethylamine lyase (encoded by the *cutCD* genes) (*Craciun and Balskus, 2012*). *S. venezuelae* lacks any gene with similarity to *cntA*, and thus does not use an equivalent pathway to generate TMA. It does possess homologues of *cutCD*; however, these genes were more highly expressed (~5 fold) in static *S. venezuelae* cultures (where no TMA was ever detected), than in spreading cultures. This suggested that these gene products may not direct TMA production in *S. venezuelae*. TMA can also be produced upon biogenic reduction of trimethylamine *N*-oxide (TMAO) by TMAO reductases. Bacteria known to carry out this reaction typically encode one or more TMAO reducase operons, including some combination of *torSTRCAD* (or *torSTRCADE*), *torYZ*, *dmsABC*, and *ynfEFGH* (*Dunn and Stabb, 2008*; *McCrindle et al., 2005*). *S. venezuelae* encodes homologs to some of these genes [specifically *torA* (top hit: SVEN_1326), *dmsAB* (top hit: SVEN_3040-3039), and *ynfEFG* (top hit: SVEN_3040, 3040 and 3039)]. In our RNA seq data, however, all of these genes (along with more divergent homologs) were expressed at extremely low levels, with equivalent levels for each gene being observed in both static and exploratory cultures. This suggested these gene products were unlikely to be involved in converting TMAO to TMA in *S. venezuelae*.

## TMA induces exploratory growth by raising the pH of the growth medium

To confirm that TMA could raise the pH of the growth medium in the same way as explorer cells, we measured the pH of non-inoculated YPD agar around dishes containing TMA, and found the pH rose from 7.0 to 9.5. To test whether TMA induced exploratory growth by raising the pH, we repeated our plate assays described in *Figure 4E*, and buffered the agar to 7.0 using 50 or 200 mM MOPS (*Figure 4F*). The pH of these plates rose to 8.0 (as opposed to 9.5 on non-buffered plates), and TMA failed to induce *S. venezuelae* exploration to the same extent as on non-buffered plates. To further validate the pH-mediated effect of TMA, we tested whether ammonia (another basic VOC) had the same effect (*Figure 4E*). After seven days, ammonia induced *S. venezuelae* exploratory growth, suggesting that VOC-mediated alkalinity stimulated *Streptomyces* exploration.

## TMA can reduce the survival of other bacteria

TMA can alter the developmental program of streptomycetes, and is known to modify the antibiotic resistance profiles of bacteria (*Letoffe et al., 2014*). Given the antibiotic production capabilities of *Streptomyces* bacteria, we wondered whether the release of TMA might also inhibit the growth of other bacteria. To explore this possibility, we set up a small petri dish of YPD agar inside a larger dish of YPD agar (*Figure 5*).*S venezuelae* and *S. cerevisiae* (exploratory cultures) or *S. venezuelae* alone (static cultures) were inoculated on the smaller dish, and plates were incubated for 10 days. The soil-dwelling bacteria *Bacillus subtilis* or *Micrococcus luteus* were then spread on the larger petri dish. Growth of *B. subtilis* and *M. luteus* in association with exploratory or static *S. venezuelae* cultures, was then assessed after overnight incubation. *B. subtilis* and *M. luteus* colony numbers were reduced by an average of 17.4% and 25.1%, respectively, on plates exposed to VOCs produced by exploratory *S. venezuelae*, relative to those grown adjacent to static cultures. We determined that the pH of medium adjacent to exploratory *S. venezuelae* had risen to 9.5, suggesting that TMA and its pH-modulatory effects could be responsible for the growth-inhibition of these bacteria.

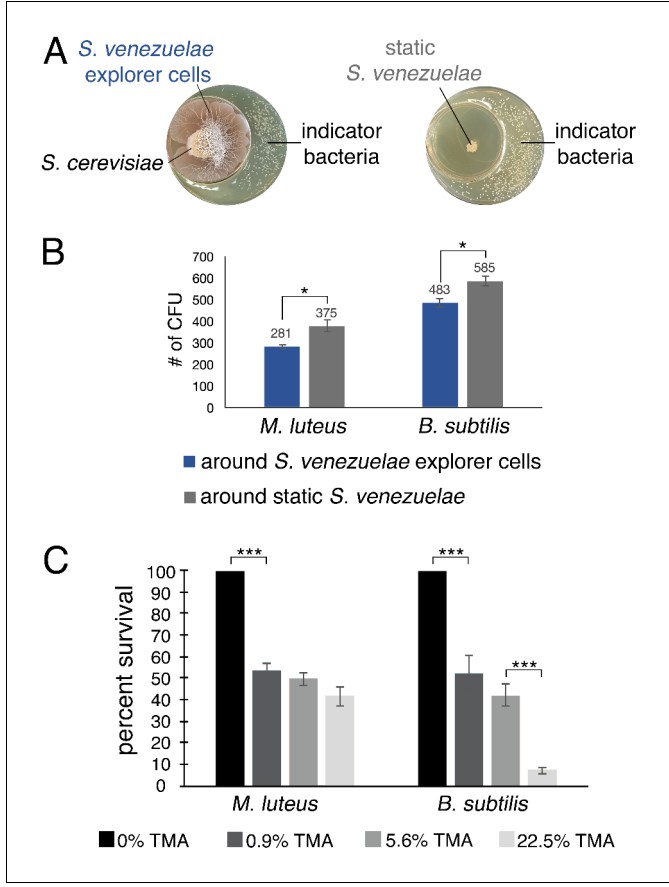

**Figure 5.** *S. venezuelae* VOCs inhibit the growth of other bacteria. (**A**) *S. venezuelae* was grown beside *S. cerevisiae* (left) or alone (right) on YPD agar in a small dish placed within a larger dish containing YPD medium. After 10 days, an indicator strain (*B. subtilis* or *M. luteus*) was spread around the dish. (**B**) Quantification of *B. subtilis* and *M. luteus* colonies following growth adjacent to static or explorer *S. venezuelae* cultures. Values represent the mean ± standard error for three replicates. The asterisk (*) indicates p<0.05, as determined by a Student's t-test. (**C**) Quantification of *B. subtilis* and *M. luteus* survival following incubation around small dishes containing TMA solutions at concentrations ranging from 0–22.5%. Plates were incubated at room temperature for two days. Percent survival indicates the $OD_{600}$ of strains around wells containing 0.9%, 5.6%, or 22.5% TMA solutions compared to the $OD_{600}$ of strains around wells containing $H_2O$ (100% survival). Values represent the mean ± standard error for three biological replicates, and each biological replicate is the average of four technical replicates. The asterisks (***) indicate p<0.005, as determined by a Student's t-test.

To directly test the inhibitory potential of TMA, we set up an equivalent assay, where the TMA-producing *S. venezuelae-S. cerevisiae* combination was substituted with aqueous TMA solutions of varying concentrations. We spread *B. subtilis*, and *M. luteus* around the TMA-containing receptacles, and after seven days, quantified growth (*Figure 5C*). We observed an approximately 50% drop in viable cells when exposed to 0.9% TMA, and in the case of *B. subtilis*, a further drop in viability was observed as TMA concentrations increased. This confirmed that TMA adversely affected the growth and survival of other soil bacteria.

## Discussion

The canonical multicellular lifecycle of *Streptomyces* bacteria begins with fungus-like hyphal growth, and ends with sporulation (*Figure 1A*). In this system, spore dispersal is the sole means by which these bacteria can establish themselves in new environments. Here, we demonstrate a new developmental behavior for *Streptomyces* that provides them with an alternative means of colonizing new habitats. In response to fungal neighbours and nutrient (glucose) depletion, *Streptomyces* can escape the confines of their classically defined lifecycle, and initiate exploratory growth. Exploratory growth is remarkably relentless: explorer cells are not limited by inanimate barriers, and can grow over abiotic surfaces. Explorer cells alter their local environment through the release of the alkaline, volatile compound TMA. Emitting TMA not only promotes exploratory behaviour by the producing cells, it also functions as an airborne signal that elicits an exploratory response in physically distant streptomycetes, and provides further fitness benefits by inhibiting the growth of other bacteria.

### Metabolic cues trigger a developmental switch

*S. venezuelae* exploration is triggered by two key metabolic cues: glucose depletion and a rise in pH. We observed exploratory growth under low glucose conditions. In low-glucose areas of the soil, *Streptomyces* may initiate exploratory growth in an attempt to colonize environments with more readily available nutrients, whereas in high-glucose areas (*e.g.* near plant roots, or in association with fruit) (*Kliewer, 1965*; *Lugtenberg et al., 1999*; *Romano and Kolter, 2005*), exploration may be less advantageous, initiating only after nearby fungi – or other microbes – consume the existing glucose supply. Microbial alteration of nutrient profiles is likely to be common in the soil environment (*e.g.* *Romano and Kolter, 2005*), and we expect that the exploratory growth away from glucose-depleted areas would provide a benefit analogous to that of motility systems in other bacteria. Although the mechanism underlying exploration remains to be elucidated, it may be linked to sliding motility given its apparently passive nature (no appendages involved), and the fact that *Streptomyces* are known surfactant producers.

*S. venezuelae* exploration is also promoted by a self-induced rise in extracellular pH. Alkaline growth conditions trigger morphological switches in a range of fungi, including the human pathogens *C. albicans*, *C. neoformans,* and *Aspergillus fumigatus* (*Bertuzzi et al., 2014*; *Davis et al., 2000*; *O'Meara et al., 2014*). This is this first time this phenomenon has been observed in bacteria.

### Volatile compounds promote communication and enhance competition

Exploratory growth by *Streptomyces* cells is coordinated by the airborne compound TMA. TMA can further induce exploration in physically distant streptomycetes. Importantly, this volatile signal is not limited to *S. venezuelae,* and can be both transmitted and sensed by other *Streptomyces* species. Consequently, it is possible for *Streptomyces* to respond to TMA produced by other bacteria and initiate exploratory growth under conditions where glucose concentrations are high and/or glucose-titrating organisms are absent. Developmental switching in response to VOC eavesdropping has not been previously reported, but exploiting community goods in this way is not unprecedented. For example, quorum signals and siderophores produced by one organism can be taken up or used by others (*Lyons and Kolter, 2015*; *Traxler et al., 2012*). The VOC repertoire of microorganisms appears to be vast (*Chuankun et al., 2004*; *Insam and Seewald, 2010*; *Kai et al., 2009*; *Schöller et al., 2002*; *Schulz and Dickschat, 2007*; *Wilkins and Schöller, 2009*). Volatile compounds have historically been implicated in the 'avoidance responses' of fungi, promoting their growth away from inanimate objects (*Cohen et al., 1975*; *Gamow and Böttger, 1982*). Increasingly, these compounds are now being found to have important roles in communication between physically separated microbes (*Audrain et al., 2015*; *Bernier et al., 2011*; *Briard et al., 2016*; *Kim et al., 2013*;

*Letoffe et al., 2014*; *Schmidt et al., 2015*, *2016*; *Tyc et al., 2015*; *Wang et al., 2013*; *Wheatley, 2002*). A range of fungi use the volatile alkaline compound ammonia to induce morphological switches in other fungi, and to mediate inhibition of neighbouring colonies (*Palková et al., 1997*). Our observations suggest that VOCs may also be key bacterial morphological determinants, communicating developmental switches both within and between different microbial species.

In addition to serving as communication signals, VOCs may also provide their producing organisms with a competitive advantage in the soil. Volatile molecules can modulate the antibiotic resistance profiles of bacteria (*Letoffe et al., 2014*), and can themselves have antifungal or antibacterial activity (*Schmidt et al., 2015*). TMA is a particularly potent example. Here, we show that exposure of other bacteria to TMA inhibits their growth, while previous work has revealed that TMA exposure increases bacterial sensitivity to aminoglycoside antibiotics. Notably, *Streptomyces* synthesize an extraordinary range of antibiotics, including many aminoglycosides. Thus in the soil, *Streptomyces*-produced TMA may have direct antibacterial activity, in addition to sensitizing bacteria to the effect of *Streptomyces*-produced antibiotics. The ability of *Streptomyces* to modulate the growth of other soil-dwelling bacteria during exploratory growth would maximize their ability to colonize new environments, and exploit whatever nutrients are present.

## Ecological implications for exploratory growth within microbial communities

Exploratory growth represents a powerful new addition to the *Streptomyces* developmental repertoire, and one that appears to be well-integrated into the existing life cycle. When grown next to yeast, explorer cells emerge from a mass of sporulating cells (*Figure 6*). This functional differentiation represents an effective bet-hedging strategy, whereby spreading explorer cells scavenge nutrients for the group, while the sporulating cells provide a highly resistant genetic repository, ensuring colony survival in the event of failed exploration. Explorer cells resemble vegetative hyphae, in that their surface is hydrophilic; however, unlike traditional vegetative hyphae, explorer cells do not appear to branch. We presume that explorer cells dispense with frequent branching as a trade-off for the ability to rapidly spread to new environments. Exploratory growth also occurs independently of the typical *bld*- and *whi*-developmental determinants, supporting the notion that this is a unique growth strategy. It is possible, however, given the slower exploration observed for *bldN* mutants (where *bldN* encodes a sigma factor), that BldN regulon members help to facilitate the exploration process.

While we observed exploratory growth in a subset of *Streptomyces* species, it is possible that this capability is more broadly conserved and is stimulated by different conditions than those investigated here. Indeed, microbes are abundant in the soil, and interactions between different organisms within these communities are likely to be more the norm than the exception. Our work illustrates the

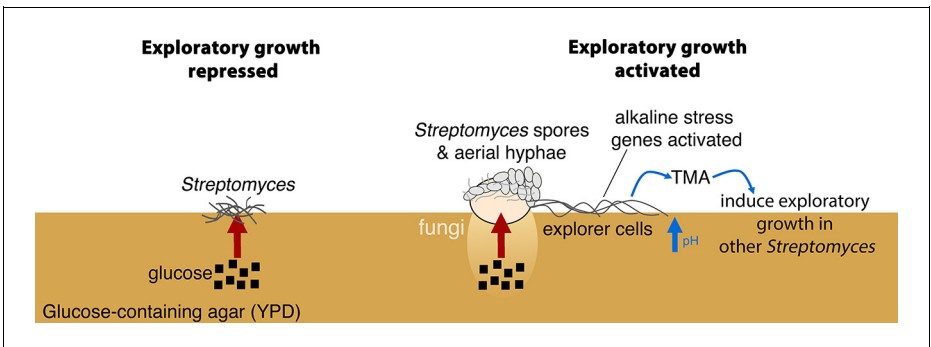

**Figure 6.** New model for *Streptomyces* development. When *S. venezuelae* is grown alone on glucose-rich medium *S. venezuelae* exploratory growth is repressed (left). When *S. venezuelae* is grown beside *S. cerevisiae* or other yeast on glucose-rich medium (right), the yeast metabolizes glucose, relieving the repression of *S. venezuelae* exploration. *S. venezuelae* explorer cells produce the volatile pheromone TMA, which raises the pH of the medium from 7.0 to 9.5. Explorer cells activate alkaline stress genes to withstand the alkaline pH. TMA, and its associated medium alkalinisation, can induce exploratory growth in physically separated *Streptomyces*.

importance of inter-species interactions in bacterial development, as a key to revealing novel growth strategies. It also emphasises the need to consider long-range communication strategies, in the form of volatile compounds, which may play widespread roles in regulating development and metabolic activities in microbial communities.

## Materials and methods

### Strains, plasmids, media and culture conditions

Strains, plasmids and primers used in this study are listed in *Supplementary file 1*. *S. venezuelae* ATCC 10712 was grown on MYM (maltose-yeast extract-malt extract) agar medium for spore stock generation. Spreading was investigated during growth on the surface of YPD (yeast extract-peptone-dextrose/glucose) agar, glucose-deficient YP (G-) agar, yeast extract agar supplemented with different amino acid sources (tryptone or 2% casamino acids) or YPD/G- agar medium supplemented with citrate, acetate, borate or MOPS buffer. All strains were grown at 30°C, apart from the TMA experiments which were conducted at room temperature in a fume hood. *S. cerevisiae* strain BY4741 (*MATa; his3Δ1; leu2Δ0 ura3Δ0 met15Δ0*) was grown on the same spreading-investigative media at 30°C or room temperature. Prior to plating *S. venezuelae* and *S. cerevisiae* together, *S. venezuelae* was cultured in liquid MYM at 30°C, while *S. cerevisiae* was grown in liquid YPD at 30°C overnight. Three microliters of *S. venezuelae* cultures were applied to the right of 3 µL *S. cerevisiae* on the surface of YPD agar medium, and plates were then incubated at 30°C or room temperature for up to 14 days

### Scanning electron microscopy (SEM) and light microscopy

SEM was used to examine strains grown on YPD or MYM agar for up to 14 days. Samples were prepared and visualized using a TEMSCAN LSU scanning electron microscopy as described previously (*Haiser et al., 2009*). To monitor the rate of exploratory growth (*Video 1*), an Olympus SZX12 Sterioscope and CoolSNAP HQ photometric camera were used to capture 70 frames of growth over the course of 17 hr.

### Phylogenetic analyses

*rpoB* (*Guo et al., 2008*) was amplified from each of the 19 exploration-competent wild isolates using primers RpoBPF and RpoBPR (*Supplementary file 1*), before being sequenced using RpoBF1 and RpoBR1 (*Supplementary file 1*). Trimmed *rpoB* sequences were aligned using Mafft version 7.2.6.6. A maximum likelihood tree was built using RAxML version 8.2.4 (*Stamatakis, 2006*), using a GTRGAMMA model of nucleotide substitution, with 500 bootstrap replicates to infer support values of nodes. Outputs were visualized using FigTree.

### Yeast library screening

Overnight cultures of *S. venezuelae* were spotted onto rectangular plates containing YPD agar (OmniTray: Nunc International) using a 384-pin replicator. Each strain of a *S. cerevisiae* BY4741 haploid deletion library was inoculated beside an individual *S. venezuelae* colony using a 384-pin replicator. Plates were grown for five days at 30°C and screened for an absence of *S. venezuelae* exploratory growth. Yeast mutants unable to stimulate *S. venezuelae* exploratory growth were retested on individual YPD agar plates. For *C. albicans* deletion screens, *C. albicans* GRACE collection tetracycline repressible deletion mutants (*Roemer et al., 2003*) were inoculated beside *S. venezuelae* on YPD agar plates. Mutants were induced using 1 or 5 µg/mL tetracycline, which is below the minimum inhibitory concentration of tetracycline for *S. venezuelae.*

### Glucose assays and measurement of pH

Measurements of glucose levels beneath *S. cerevisiae* colonies and in YPD alone were performed using a Glucose (GO) Assay Kit (Sigma). For all experiments, pH levels of solid agar were measured using one or a combination of pH sticks and the pH indicator dye bromothymol blue (Sigma, St Louis, MO).

## Chemical mutagenesis and whole-genome sequencing

Approximately $10^8$ *S. venezuelae* spores were added to 1.5 mL 0.01 M KPO$_4$ at pH 7.0. Spores were centrifuged and resuspended in 1.5 mL 0.01 M KPO$_4$ at pH 7.0. The spores were then divided into two 750 µL aliquots in screw-cap tubes. As a control, 25 µL H$_2$O was added to one aliquot, while 25 µL ethyl methanesulfonate (EMS, Sigma, M0880) was added to the other aliquot. Tubes were vortexed for 30 s, and incubated shaking at 30°C for 1 hr, with an additional inversion being performed every 10 min. Spores were centrifuged at 3381 $\times g$ for 3 min at room temperature, prior to being resuspended in 1 mL freshly made and filter-sterilized 5% w/v sodium thiosulfate solution. Spores were washed twice in 1 mL H$_2$O, after which they were resuspended in 1 mL H$_2$O. For each tube, a dilution series ranging from $10^{-4}$ to $10^{-8}$ was made using H$_2$O, and 100 µL of each dilution was then spread onto MYM agar plates and incubated for three days at 30°C. Individual colonies were counted to ensure that survival of the EMS-treated spores was, at most, 50% that of the untreated (H$_2$O) control. Colonies were collected from plates inoculated with EMS-treated spores, and were screened for loss of spreading capabilities on G- agar plates. Select mutants were then tested for their inability to spread when plated next to yeast; those mutants that also failed to initiate spreading in the presence of *S. cerevisiae* were grown in liquid MYM, and chromosomal DNA was extracted using the Norgen Biotek Bacterial Genomic DNA Isolation kit for downstream sequencing.

Using the Illumina Nextera XT DNA sample preparation kit, DNA libraries were prepared for three non-exploratory *S. venezuelae* mutants, alongside their wild type *S. venezuelae* parent. Whole genome-sequencing was performed on an Illumina MiSeq instrument (Illumina, San Diego, CA, USA) using 150 bp paired-ends reads. Reads were aligned to the *S. venezuelae* reference genome using Bowtie 2 (*Langmead and Salzberg, 2012*) and were converted to BAM files using SAMtools (*Li et al., 2009*). Single nucleotide polymorphisms (SNPs) were called using SAMtools mpileup and bcftools, and SNP locations, read depth, and identities were generated using VCFtools (*Danecek et al., 2011*).

## Construction of *cydCD* (cytochrome bd oxidase) deletion strain and mutant complementation

An in-frame deletion of *sven_3715-3716* was generated using ReDirect technology (*Gust et al., 2003*). The coding sequence was replaced by an *oriT*-containing apramycin resistance cassette. The gene deletion was verified by PCR, using combinations of primers located upstream, downstream and internal to the deleted genes (see *Supplementary file 1*). The *cydCD* mutant phenotype was complemented using a DNA fragment encompassing the WT genes, *sven_3713-3714*, and associated upstream and downstream sequences (see *Supplementary file 1*), cloned into the integrating plasmid vector pSET152. To control for any phenotypic effects caused by plasmid integration, pSET152 alone was introduced into wild type and the *cydCD* mutant strains, and these strains were used for phenotypic comparison with the complemented mutant strain.

## RNA isolation, library preparation and cDNA sequencing

RNA was isolated as described previously from two replicates of *S. venezuelae* explorer cells growing beside *S. cerevisiae* for 14 days, and two replicates of *S. venezuelae* alone grown for 24 hr on YPD agar plates (we were unable to isolate high quality RNA from *S. venezuelae* alone at later time points). For all four replicates, ribosomal RNA (rRNA) was depleted using a Ribo-zero rRNA depletion kit. cDNA and Illumina library preparation were performed using a NEBnext Ultra Directional Library Kit, followed by sequencing using unpaired-end 80 base-pair reads using the HiSeq platform. Reads were aligned to the *S. venezuelae* genome using Bowtie 2 (*Langmead and Salzberg, 2012*), then sorted, indexed, and converted to BAM format using SAMtools (*Li et al., 2009*). BAM files were visualized using Integrated Genomics Viewer (*Robinson, 2011*), and normalization of transcript levels and analyses of differential transcript levels were conducted using Rockhopper (*McClure et al., 2013*). RNA-seq data has been deposited in NCBI's Gene Expression Omnibus and are accessible through GEO Series accession number GSE86378 (http://www.ncbi.nlm.nih.gov/geo/query.acc.cgi?token=idmrgcmexranpun&acc=GSE86378).

## Analysis of volatile metabolites via GC×GC-TOFMS

*S. venezuelae* and WAC0566 were grown in liquid YPD (G+) or YP (G-) for three days. For each strain and condition, six biological replicates were grown, and for each, three technical replicates were analyzed. Four milliliters of each culture supernatant were transferred to 20 mL air-tight headspace vials, which were stored at −20°C prior to volatile analysis. Headspace volatiles were concentrated on a 2 cm triphasic Divinylbenzene/Carboxen/Polydimethylsiloxane (DVB/CAR/PDMS) solid-phase microextraction (SPME) fiber (Supelco, Bellenfonte, PA) (30 min, 50°C, 250 rpm shaking). Volatile molecules were separated, identified, and relatively quantified using two-dimensional gas chromatography time-of-flight mass spectrometry (GC×GC-TOFMS), as described previously (*Bean et al., 2012*; *Rees et al., 2016*). The GC×GC-TOFMS (Pegasus 4D, LECO Corporation, St. Joseph, MI) was equipped with a rail autosampler (MPS, Gerstel, Linthicum Heights, MD) and fitted with a two-dimensional column set consisting of an Rxi−624Sil (60 m × 250 µm×1.4 µm (length × internal diameter × film thickness); Restek, Bellefonte, PA) first column followed by a Stabilwax (Crossbond Carbowax polyethylene glycol; 1 m × 250 µm×0.5 µm; Restek, Bellefonte, PA) second column. The main oven containing column one was held at 35°C for 0.5 min, and then ramped at 3.5 °C/min from 35°C to 230°C. The secondary oven containing column 2, and the quad-jet modulator (2 s modulation period, 0.5 s alternating hot and cold pulses), were heated in step with the primary oven with +5°C and +25°C offset relative to the primary oven, respectively. The helium carrier gas flow rate was 2 mL/min. Mass spectra were acquired over the range of 30 to 500 a.m.u., with an acquisition rate of 200 spectra/s. Data acquisition and analysis was performed using ChromaTOF software, version 4.50 (LECO Corp.).

## Identification of candidate volatile signals

Chromatographic data was processed and aligned using ChromaTOF. For peak identification, a signal-to-noise (S/N) cutoff was set at 100, and resulting peaks were identified by a forward search of the NIST 2011 Mass Spectral Library. For the alignment of peaks across chromatograms, maximum first and second-dimension retention time deviations were set at 6 s and 0.15 s, respectively, and the inter-chromatogram spectral match threshold was set at 600. Analytes that were detected in greater than half of exploration-promoting *Streptomyces* cultures (grown in G- medium) and not detected in media controls or *S. venezuelae* grown in G+ medium (failed to promote exploration), were considered candidate molecules associated with the phenotype of interest.

## Assays for volatile-mediated phenotypes

Aqueous solutions (1.5 mL) of commercially available TMA solutions (Sigma), ammonia solutions (Sigma) or water (negative control) were added to small, sterile plastic containers and placed in a petri dish containing 50 mL YPD agar. TMA solutions were typically diluted to 11.5% w/v, although concentrations as low as 0.9% were able to promote spreading and inhibit the growth of other bacteria. Ammonia solutions of 0.1–1 M were used, and all were able to induce spreading. *S. venezuelae* was inoculated around the small vessels, after which the large petri dish was closed and incubated in the fume hood at room temperature for up to 10 days. For buffering experiments, YPD plates were supplemented with 50 or 200 mM MOPS buffer (pH 7.0). Medium pH was measured as above, while colony surface areas were measured using ImageJ (*Abràmoff et al., 2004*). For bacterial survival assays around TMA-containing vessels, *B. subtilis* and *M. luteus* strains were grown overnight in LB medium, before being subcultured to an $OD_{600}$ of 0.8. One hundred microliters of each culture were then spread on YPD agar plates, adjacent to water or TMA-containing vessels. For assays to measure how *S. venezuelae* explorer VOCs affect the survival of other bacteria, *S. venezuelae* was grown alone or beside *S. cerevisiae* in a small petri dish containing YPD agar. This small dish was placed inside a larger dish containing YPD agar. Plates were grown for 10 days, before *B. subtilis* and *M. luteus* were subcultured to an $OD_{600}$ of 0.8, and diluted 1/10 000. Fifty microliters of each culture were then spread on the larger plate containing YPD agar, and colonies were quantified after overnight growth.To test the effect of TMA on *B. subtilis* and *M. luteus* growth, these indicator strains were grown overnight in LB medium, before being subcultured to an $OD_{600}$ of 0.8. One hundred microlitres were spread around wells containing 1.5 mL solutions of TMA at different concentrations on YPD (water control, 0.9%, 5.6% and 22.5%). Plates were incubated for two days at room temperature in the fume hood, before cells were scraped into 2 mL YPD and vigorously mixed. Dilution

series were used to measure the $OD_{600}$ of the resulting cell suspensions. Error bars indicate standard error of three biological replicates, and four technical replicates of each.

## Acknowledgements

We would like to extend our thanks to Dr. Leah Cowen and Teresa O'Meara for access to the *S. cerevisiae* and GRACE yeast library collections, Dr. J.P Xu and Aaron Vogen for access to environmental yeast strains, Dr. Gerry Wright for access to his wild *Streptomyces* library, Dr. Mark Buttner and Maureen Bibb for the *S. venezuelae* developmental mutants, Dr. Chris Yip and Amine Driouchi for microscopy assistance, Ben Furman for assistance with the *Streptomyces* phylogeny, David Crisante for artistic assistance, and Dr. Heather Bean, Dr. Mark Buttner, Dr. Erin Carlson, Andy Johnson and Matt Moody for helpful discussions.

## Additional information

### Funding

| Funder | Grant reference number | Author |
| --- | --- | --- |
| Natural Sciences and Engineering Research Council of Canada | Vanier Scholarship | Stephanie E Jones |
| Ontario Government | Ontario Graduate Scholarship | Louis Ho |
| Canadian Institutes of Health Research | MOP133636 | Justin R Nodwell |
| Natural Sciences and Engineering Research Council of Canada | RGPIN-2015-04681 | Marie A Elliot |

The funders had no role in study design, data collection and interpretation, or the decision to submit the work for publication.

### Author contributions

SEJ, K., CAR, Conception and design, Acquisition of data, Analysis and interpretation of data, Drafting or revising the article; JEH, JRN, MAE, Conception and design, Analysis and interpretation of data, Drafting or revising the article

## Additional files

### Supplementary files

• Supplementary file 1. Supplementary tables. (a) VOCs identified using GC×GC-TOFMS. (b) Effects of media composition on *S. venezuelae* exploration when grown in the absence of yeast. (c) Oligonucleotides used in this study.

### Major datasets

The following dataset was generated:

| Author(s) | Year | Dataset title | Dataset URL | Database, license, and accessibility information |
| --- | --- | --- | --- | --- |
| Stephanie E Jones, Marie A Elliot | 2016 | Streptomyces exploration is triggered by fungal interactions and volatile signals | https://www.ncbi.nlm.nih.gov/geo/query/acc.cgi?acc=GSE86378 | Publicly available at NCBI Gene Expression Omnibus (accession no: GSE86378) |

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
