## [Decision Letter]

Thank you for submitting your article "*Streptomyces* exploration is triggered by fungal interactions and volatile signals" for consideration by *eLife*. Your article has been favorably evaluated by Richard Losick (Senior Editor) and three reviewers, one of whom is a member of our Board of Reviewing Editors. The following individuals involved in review of your submission have agreed to reveal their identity: Jean Marc Ghigo (Reviewer #2); Matthew Traxler (Reviewer #3).

The reviewers have discussed the reviews with one another and the Reviewing Editor has drafted this decision to help you prepare a revised submission.

This paper by Jones and colleagues is a fascinating report of a new type *Streptomyces* differentiation induced by the presence of neighboring fungi. Using elegant experimental designs the authors show that rapid glucose consumption by the fungus turns on a deep *Streptomyces* metabolic reprogramming, alkalinization of the surrounding environment and a switch to a new "exploratory" lifestyle. The pH drop seems to be driven by a volatile molecule, TMA, which raises the interesting possibility that *Streptomyces* colonies can exchange differentiation signals at a distance, and possibly use TMA from other bacteria to adapt their lifecycle.

Overall, the paper is written clearly and concisely. The experiments are well performed and controlled. Nevertheless, the paper should be improved in several instances to be suitable for publication.

The demonstration that pH alone can induce the exploratory behavior appears to be missing. Can *Streptomyces* switch to exploratory mode simply if the plates are buffered to pH 9.5? The authors show that the TMA effects are reduced when buffered by MOPS but they their model suggests that lowering the pH of the plates alone should be sufficient to induce exploratory growth.

Is the *cydCD* mutant capable of inducing exploration in the WT in the VOC assay? This would speak to an important question: Does the *cydCD* mutant fail to respond to alkaline conditions or does it fail to make them?

The authors conclude that none of the *bld* mutants showed altered exploratory phenotypes. Is it really so? It looks like the *bldN* mutant did not spread as much as the WT. Did it later spread like the WT? Could the exploratory cells at least partially require *bldN* to achieve this exploratory morphology?

The study is somewhat vague regarding the origin of TMA. The authors indicate the lack of carnitine oxygenase able to convert L-carnitine into TMA. In many other organisms, TMA is produced upon biogenic reduction of trimethylamine oxide (TMAO). Does *S. venezuelae* possess TMAO reductase? If so, does *S. venezualae* medium contains TMAO? In addition, in absence of a TMA-producing mutant, it is not known whether TMA is the sole active molecule. For example, ammonia is shown to produce similar phenotypes. Can the authors exclude other molecules?

Elucidating the molecular and cellular basis of the exploratory lifestyle is not within the immediate scope of this paper but the process itself is very little discussed throughout the paper. Is motility involved? Or is it only growth and why in this case the exceptional spreading? The cells are mentioned to be hydrophilic, is the overall moisture of the colony increased? Is a surfactant involved? Leads into these questions could come from more analysis/description of the 48 mutants that were isolated. Also, why do the authors think that mutants that cannot spread in Glucose-deprived medium can still do so in the presence of yeast?

The work performed by Zdena Palková (https://www.ncbi.nlm.nih.gov/pubmed/?term=Zdena+Palková++ammonia) on how ammonia-dependent medium alkalization induce morphogenetic switch in yeast should at least be mentioned: it is after all the best studied alkaline volatile compounds triggering a morphogenetic switch.

In the first paragraph of the subsection “The yeast TCA cycle must be intact to stimulate *S. venezuelae* exploratory behaviour” and Figure 2: The text mentions nine genes affected in TCA cycle => the figure (diagram part of the figure) only shows eight of them. However, CYT1, involved in oxaloacetate to citrate transition (TCA cycle part of the figure) is not shown: is it the missing one?

---

## [Author Response]

[…]

*Overall, the paper is written clearly and concisely. The experiments are well performed and controlled. Nevertheless, the paper should be improved in several instances to be suitable for publication.*

*The demonstration that pH alone can induce the exploratory behavior appears to be missing. Can Streptomyces switch to exploratory mode simply if the plates are buffered to pH 9.5? The authors show that the TMA effects are reduced when buffered by MOPS but they their model suggests that lowering the pH of the plates alone should be sufficient to induce exploratory growth.*

To address whether raising the pH of the medium was sufficient to induce exploratory growth, we buffered plates containing YPD medium to pH 9.0 using 0 (control), 50, 100 or 200 mM borate. When *S. venezuelae* alone was grown on each plate, we found it grew slowly on all borate-buffered plates, and did not initiate exploration under any condition. These results suggest that high pH alone cannot induce exploratory behavior.

When *S. venezuelae* grows beside yeast on YPD, or alone on YP, cells first grow at a neutral pH of 7.0, before transitioning to exploratory growth, when the medium pH rises to 9.5. Given that *S. venezuelae* is unable to explore when grown on plates buffered to pH 9.0, it appears that exploratory growth requires an adaptation phase, during the transition to alkaline conditions.

We have added a figure to the Supplementary Information (Figure 3—figure supplement 2), showing the borate experiments, and discuss the observation that high pH alone cannot stimulate exploration (subsection “Exploration is glucose-repressible and pH-dependent”, last paragraph).

*Is the cydCD mutant capable of inducing exploration in the WT in the VOC assay? This would speak to an important question: Does the cydCD mutant fail to respond to alkaline conditions or does it fail to make them?*

We have conducted a series of experiments aimed at determining whether the *cydCD* mutant fails to respond to alkaline conditions or fails to generate an alkaline environment:

To address whether the *cydCD* mutant could respond to alkaline conditions, we used our two-compartment assay. Wild type *S. venezuelae* was inoculated beside *S. cerevisiae* on YPD (exploration-promoting conditions), and the *cydCD* mutant was inoculated alone on YPD in the adjacent compartment (exploration deficient conditions, except when stimulated by an alkaline VOC). After ten days, we found the mutant strain was able to explore in response to VOCs produced by the wild type strain. We have now included these results as Figure 4—figure supplement 1, and describe this experiment in the first paragraph of the subsection “S. venezuelae exploratory cells use VOCs to induce exploration in other *streptomycetes* at a distance”. Additionally, we placed TMA solutions in a small plastic container in the centre of a YPD agar plate, and found exploration was induced in the *cydCD* strain, although exploration progressed more slowly than for the wild type strain (please see Figure 7).

Author response image 1.**DOI:**
http://dx.doi.org/10.7554/eLife.21738.022

Collectively, these experiments show that the *cydCD* mutant can explore in response to alkaline conditions.

To determine whether the *cydCD* mutant could produce an alkaline VOC, we set up a two-compartment assay with *cydCD* inoculated beside *S. cerevisiae* on YPD, and wild type *S. venezuelae* alone in the adjacent YPD compartment. Consistent with our previous observations, the *cydCD* mutant beside yeast did not explore, and wild type *S. venezuelae* in the adjacent compartment was not induced to explore. This suggested that either the *cydCD* mutant did not produce an alkaline VOC, or it was not produced at sufficient levels under these conditions to influence the growth of the wild type strain in the adjacent compartment.

*The authors conclude that none of the bld mutants showed altered exploratory phenotypes. Is it really so? It looks like the bldN mutant did not spread as much as the WT. Did it later spread like the WT? Could the exploratory cells at least partially require bldN to achieve this exploratory morphology?*

Although the *bldN* mutant did not spread as much as the wild type strain by day 14, it did eventually cover the entire surface of the agar plate in the same way as wild type. Similarly, when the *bldN* mutant strain demonstrated exploratory behavior on other media types – e.g.alone on YP – it did not spread as quickly as wild type, but did eventually colonize the entire agar surface. Thus *bldN* is not necessary for *S. venezuelae* exploratory growth; however, we cannot rule out the possibility that BldN regulon members may contribute to this process. We have now noted the slower spreading of the *bldN* mutant in our Results section (subsection “Physical association with yeast stimulates *Streptomyces* exploratory behaviour”, third paragraph), and address a possible role for BldN regulon members in facilitating rapid exploration in our Discussion (subsection “Ecological implications for exploratory growth within microbial communities”, first paragraph).

*The study is somewhat vague regarding the origin of TMA. The authors indicate the lack of carnitine oxygenase able to convert L-carnitine into TMA. In many other organisms, TMA is produced upon biogenic reduction of trimethylamine oxide (TMAO). Does S. venezuelae possess TMAO reductase? If so, does S. venezualae medium contains TMAO? In addition, in absence of a TMA-producing mutant, it is not known whether TMA is the sole active molecule. For example, ammonia is shown to produce similar phenotypes. Can the authors exclude other molecules?*

The regulation and expression of TMAO reductases have been studied in a number of genera, including *Escherichia, Shewanella, Rhodobacter*, and *Vibrio* (e.g.Dunn and Stabb, 2008, McCrindle et al., 2005). These bacteria typically have two or three TMAO reductase operons, including some combination of *torSTRCAD* (or *torSTRCADE), torYZ, dmsABC*, and *ynfEFGH*. TMAO reductases have not been studied in *Streptomyces* or other actinomycetes. We used BLAST to identify TMAO reductase homologs of all known TMAO reductase genes from other bacteria in *S. venezuelae*. We identified several genes in *S. venezuelae* that were homologous to genes in known TMAO reductase operons [specifically *torA* (top hit: *SVEN_1326), dmsAB* (top hit: *SVEN_3040-3039),* and *ynfEFG* (top hit: *SVEN_3040, 3040 and 3039)*]. We examined the expression of these genes using our RNA-seq data, and found that all of them (along with more divergent homologs) were expressed at extremely low levels. Furthermore, equivalent transcript levels for each gene were observed in both static and exploratory cultures. This suggests these gene products were unlikely to be involved in converting TMAO to TMA in *S. venezuelae*.

We agree entirely that the conversion of TMAO to TMA is an important means by which TMA can be produced, and we have added this point, and the information on *S. venezuelae* TMAO homologs to the manuscript (subsection “The VOC trimethylamine stimulates *Streptomyces* exploratory behaviour”, last paragraph).

At this stage, we do not have a TMA production mutant, as TMA production – either from quaternary amines or TMAO – is not well understood in bacteria, and has not been studied in the actinomycetes. While we acknowledge that without such a mutant, it is not possible to definitively conclude that TMA is the exploration-promoting compound, our results provide three strong lines of evidence in support of this proposal:

1) TMA was one of the most abundant molecules in our GC×GC-TOFMS exploratory cultures. It was also present in *all* explorer cultures, and was completely absent from static cultures. We did not identify any other compound that was absent from static cultures and was highly abundant in the majority of exploratory cultures. We also did not identify any other highly abundant compounds with the ability to raise the pH of growth medium.

2) Exploration is associated with an increase in pH to 9.5; TMA raises the pH of the medium to a near identical level.

3) Commercially available TMA could induce *S. venezuelae* exploration in a manner that was virtually indistinguishable from that of *S. venezuelae*+yeast, indicating that TMA on its own could induce exploratory growth.

*Elucidating the molecular and cellular basis of the exploratory lifestyle is not within the immediate scope of this paper but the process itself is very little discussed throughout the paper. Is motility involved? Or is it only growth and why in this case the exceptional spreading? The cells are mentioned to be hydrophilic, is the overall moisture of the colony increased? Is a surfactant involved? Leads into these questions could come from more analysis/description of the 48 mutants that were isolated. Also, why do the authors think that mutants that cannot spread in Glucose-deprived medium can still do so in the presence of yeast?*

The exploratory phenotype appears to be most similar to sliding motility, where sliding motility is defined as a form of passive surface motility powered by growth and aided by a surfactant. We have preliminary evidence suggesting a surfactant is involved: the surface of the cells is hydrophilic (clarified now in subsection “Ecological implications for exploratory growth within microbial communities”, first paragraph), but underneath, the cells appear to secrete a hydrophobic/amphipathic substance. We have not yet been able to identify this compound, and we are hesitant to label exploration as sliding motility in the absence of surfactant characterization.

We isolated 45 mutants unable to spread on YP, but able to spread on YPD beside *S. cerevisiae* (along with three that failed to explore under either condition). This suggested to us that the two spreading processes may be genetically separable. Supporting this proposal (but not described in this manuscript), is our observation that a *crp* mutant has the reciprocal characteristics: it is unable to spread on YPD beside *S. cerevisiae,* but can explore when grown on YP medium. We have recently isolated a new mutant that fails to spread on both YPD with *S. cerevisiae*, and on YP, but is able to spread on a more traditional *S. venezuelae* medium MYM (maltose, yeast extract, malt extract) supplemented with peptone (a condition where wild type also explores). It therefore appears that there are diverse metabolic cues that are sensed in the lead-up to exploration (beyond simply glucose and pH), and it is likely that these mutants differ in their ability to sense/respond to these different nutritional conditions. We are currently working to tease apart the nutritional aspects in more detail, and to understand the genetic responses to these different media conditions.

*The work performed by Zdena Palková (https://www.ncbi.nlm.nih.gov/pubmed/?term=Zdena+Palková++ammonia) on how ammonia-dependent medium alkalization induce morphogenetic switch in yeast should at least be mentioned: it is after all the best studied alkaline volatile compounds triggering a morphogenetic switch.*

We have now included the findings of Palková in our discussion on VOC-induced morphogenetic switches in bacteria (subsection “Volatile compounds promote communication and enhance competition”, first paragraph).

*In the first paragraph of the subsection “The yeast TCA cycle must be intact to stimulate S. venezuelae exploratory behaviour” and Figure 2: The text mentions nine genes affected in TCA cycle => the figure (diagram part of the figure) only shows eight of them. However, CYT1, involved in oxaloacetate to citrate transition (TCA cycle part of the figure) is not shown: is it the missing one?*

The CIT1 mutant induces *S. venezuelae* exploration (it is only after citrate production that we observe this defect). We have changed the text to mention eight genes affected in the TCA cycle (subsection “The yeast TCA cycle must be intact to stimulate *S. venezuelae* exploratory behaviour”, first paragraph).